# ConstrainPrompt: Code-Based Assurance of Prompt-Defined Constraints

## Abstract

Large language models (LLMs) are increasingly used in applications where outputs must satisfy hard, application–critical constraints (e.g., JSON format, lexical inclusion, and length limits). When these constraints are violated, downstream parsers may fail (e.g., invalid JSON), application behavior can become incorrect or unsafe (e.g., missing required strings or forbidden terms), and automation pipelines may halt. Although controlled text generation can mitigate violations, LLM outputs still frequently breach constraints. Therefore, post-generation evaluation is essential. Common evaluators implemented by LLM-as-a-judge or rule-based scripts under-penalize hard errors and lack robust, fine-grained evaluation control flow. We propose *ConstrainPrompt*, a verification pipeline that induces semantics-agnostic, code-verifiable constraints from natural-language prompts and compiles them into executable validators. Our method extracts code-verifiable constraints from the prompt, synthesizes a logical evaluation tree that orders global-to-local checks and resolves conditional guards, and finally generates code to validate LLM outputs. On a corpus of real-world prompts paired with LLM outputs, ConstrainPrompt improves Constraint Compliance Accuracy by 24.3% and Violation Rationale by 40.8% over an LLM-as-a-judge baseline across three models.

## 1 Introduction

Large language models (LLMs) can produce fluent and contextually appropriate text. Yet in real-world applications, semantic correctness alone is not sufficient. Agentic pipelines often chain multiple LLM calls with tools, and each step expects the previous output to satisfy concrete *output constraints* such as style, format, length limits, required fields, and lexical inclusion or exclusion (Zhou et al., 2023; Han et al., 2024). In prompts, these requirements are commonly grouped into *soft control* and *hard control* constraints (Liang et al., 2024). Soft constraints regulate properties like tone and topic: they matter for safety, user trust, and policy adherence (red in Figure 1). Hard constraints govern concrete elements of the output, such as structure, vocabulary, and numeric or length limits (blue in Figure 1). When hard constraints are violated, downstream components may fail to parse or trust the output, triggering cascading errors or halting the pipeline. Accordingly, rigorous quality assurance of LLM outputs is essential for reliability and safety. In these settings, *constraint compliance* is as critical as semantic correctness.

> Please analyze the following dialogue and evaluate it based on the criteria provided. Assign a score from 1 to 5 for each category. Scores should meaningfully distinguish quality levels. Please present your evaluation into the following JSON format:
> { "Relevance": ___ , "Completeness": ___ , "Correctness": ___ }
> Here is the dialog: {dialog}

Figure 1: A prompt example with soft and hard constraints.

A natural approach is to impose constraints during generation. Controlled text generation steers models toward application-level requirements, including keyword inclusion or exclusion, tone or style, length or numeric ranges, and machine-usable structures (e.g., JSON schemas and field completeness) (Zhang et al., 2023). These are not aesthetic niceties: they are preconditions for reliable parsing, storage, compliance, and safe execution. Despite substantial progress, control during generation reduces the incidence of violations but cannot eliminate them, especially under diverse, conditional, and task-specific constraints that arise in real prompts. Post-hoc verification is therefore necessary. Existing evaluation typically relies on one of three options (Liang et al., 2024): (i) human

evaluation, which can produce high quality judgments but is expensive, time-consuming, hard to reproduce, and prone to inconsistency (Liu et al., 2022); (ii) rule-based scripts, which are fast and low-cost for hard constraints but often operate at a single granularity (the whole output), making it difficult to express field or item level checks and conditional dependencies (Meng et al., 2022); and (iii) LLM-as-a-judge, which tends to under-penalize hard, semantics-agnostic constraints such as format, length, and required tokens, sometimes mirroring the same mistakes as the generator (e.g., miscounting or overlooking numeric limits) (Chen et al., 2024).

Compliance with hard control constraints is directly tied to application correctness: violating required structure, ranges, fields, or lexical rules undermines parsing and trust, triggering cascading failures or halting pipelines. Yet evaluation of hard constraints has received less attention than soft control, even though practitioners depend on it as the first acceptance check in production. Moreover, as discussed above, existing automated methods for hard-constraint evaluation have clear limitations. Therefore, we focus on evaluating hard control constraints and address this gap with *ConstrainPrompt*, a verification pipeline that separates compliance from semantic quality. *ConstrainPrompt* first induces from a natural-language prompt the set of semantics-agnostic, code-verifiable constraints (structural, quantitative, and lexical), together with their conditionality (whether triggered by specific input use case or always applicable). We then introduce a logical evaluation tree as an intermediate representation. It specifies the order of checks, their application scope, and input-side guards that enable a check only when its condition holds. This captures precedences (e.g., parse structure before validating fields and ranges) and yields a canonical control flow that prevents contradictory checks. Finally, we compile this tree into an executable validator. At runtime, the system executes the generated code in a sandbox environment and returns deterministic pass/fail outcomes together with human-readable reasons and a reference to the defining span in the prompt. We evaluate the pipeline on a dataset of real prompts paired with model outputs, reporting evaluation accuracy, along with ablations that isolate the contribution of the tree-guided control flow.

Our main contributions are:

- A dataset of real prompts paired with model outputs and annotated violations of prompt-defined hard constraints, and an empirical study that summarizes common constraint types in prompts.
- *ConstrainPrompt*, a method that induces code-verifiable constraints from prompts, synthesizes an evaluation tree to guide ordering and scope, and compiles the result into executable validators.
- An experimental evaluation on three models showing improvements of **24.3%** in *Constraint Compliance Accuracy* and **40.8%** in *Violation Rationale* over baseline methods.

## 2 TAXONOMY OF PROMPT CONSTRAINTS

To evaluate whether LLM outputs comply with prompt–specified constraints, we first characterize the kinds of constraints that appear in real-world prompts. We conduct an empirical analysis to derive a working taxonomy and to estimate the prevalence of each category.

**Data collection.** We build our corpus on PromptSet (Pister et al., 2024), a collection of prompts extracted from LLM-based applications (LLMapps) in open-source GitHub projects. These projects span a wide range of use cases and adoption levels from personal demos to widely deployed systems, yielding substantial variability in prompt quality. To better reflect mature practice, we focus on widely used prompt templates (predefined structures that combine static text with dynamic placeholders to create adaptable prompts for LLMapps) in these apps (Schulhoff et al., 2024; Zhao et al., 2025). To improve data quality, we follow the cleaning pipeline of Mao et al. (2025), filtering by factors such as repository popularity and prompt length.

To identify constraints within prompts, we segment each prompt into seven components following Mao et al. (2025). We then merge the original *Constraint* and *Output format* components into a single *Constraint* category and retain prompts that contain at least one such component. This procedure yields a subset of 1,232 prompts that explicitly specify constraints.

**Constraint taxonomy induction.** To characterize the constraint types that commonly appear in real-world prompts, we adopt a hybrid deductive–inductive coding protocol (Saldaña, 2021). We initialize a seed codebook from prior taxonomies (Zhang et al., 2023; Zhou et al., 2023) and then

| Category | Subcategory | Brief description | Frequency |
|----------|-------------|-------------------|-----------|
| Structural | Specific format | Output must conform to a specific data format | 26.1% |
| Quantitative | Numerical | Restrictions on output length, counts, or numerical ranges | 8.9% |
| Lexical | Lexical matching | Output must contain, match, or adhere to a specific string pattern | 13.3% |
| | Lexical exclusion | Certain words, phrases, or character patterns are prohibited in the output | 11.2% |
| Semantic | Semantic inclusion | Output must semantically include certain concepts, entities, or topics | 25.8% |
| | Semantic exclusion | Output must not semantically mention certain concepts, entities, or topics | 7.4% |
| | Qualitative | The output must exhibit specific qualities or styles | 7.3% |

Table 1: Distribution of constraint categories.

perform open coding with constant comparison on a random sample of 300 prompts. Coding is conducted by two professional annotators with more than two years of experience in prompt writing and refinement. For each prompt, annotators assign an existing category when applicable. Otherwise, a new category is proposed together with a concise definition and a canonical example (examples in Appendix A). Disagreements are resolved through discussion to produce a single adjudicated label. After each revision, the sample is recoded to verify that category boundaries remain stable. This process yields seven leaf categories grouped into four higher-level families, summarized in Table 1.

Using the finalized taxonomy, we employ GPT-4o-2024-11-20 (Hurst et al., 2024) to label all constraints in the full dataset. To assess labeling accuracy, annotators label a random sample of 100 prompts to create gold labels, which we compare against the GPT-4o outputs. On this set, GPT-4o achieves 94% accuracy. Applying the taxonomy to the entire corpus, we find that *specific format*, *numerical*, *lexical matching*, and *lexical exclusion* (the semantics-agnostic, code-verifiable families) together account for 59.5% of all constraints (Table 1), indicating that code-verifiable requirements constitute a large share of constraints in practice.

Within the code-verifiable families, *specific format* typically constrains the global structure of output, whereas the other three families apply to local content. We study how often these global and local constraints co-occur. Let $S$ be the set of prompts that include a specific-format constraint. Within $S$, 58.1% of prompts also include at least one local constraint from {numerical (N), lexical matching (M), lexical exclusion (E)}. Beyond this marginal, we observe nontrivial higher-order co-occurrence: a substantial fraction contains two or more local families ($\Pr(K \geq 2 \mid S) = 17.2\%$, with $K \in \{0, 1, 2, 3\}$), and a nonnegligible subset contains all three ($\Pr(N \cap M \cap E \mid S) = 3.9\%$). The average number of local families per prompt within $S$ is $\bar{K} = 0.79$. These statistics indicate that global structural constraints are often accompanied by field-level and token-level constraints.

These findings have two implications for evaluation. First, hierarchical constraints are common in practice, so reliable evaluation should separate global parsing from local checks and respect a coarse-to-fine order. Second, because multiple local families frequently co-occur, a verifier must support mixed constraints at different granularities and compose them deterministically. This motivates our pipeline design, which introduces an evaluation tree to enforce a coarse-to-fine order, parsing the global structure first and then applying field-level and item-level constraints via explicit branches. The tree serves as an explicit ordering guide for the validator code generator.

## 3 METHOD

We implement ConstrainPrompt, a constraint verifier that determines whether an LLM output satisfies the set of semantics-agnostic, programmatically checkable constraints specified by a prompt, covering the code-verifiable families. As shown in Figure 2, the pipeline has three stages: (i) *code-verifiable constraint extraction*, which identifies all constraints in a natural-language prompt and then filters to those that are code-verifiable; (ii) *evaluation tree synthesis*, which orders these constraints and assigns scopes to induce a canonical coarse-to-fine evaluation control flow; and (iii) *evaluation code*

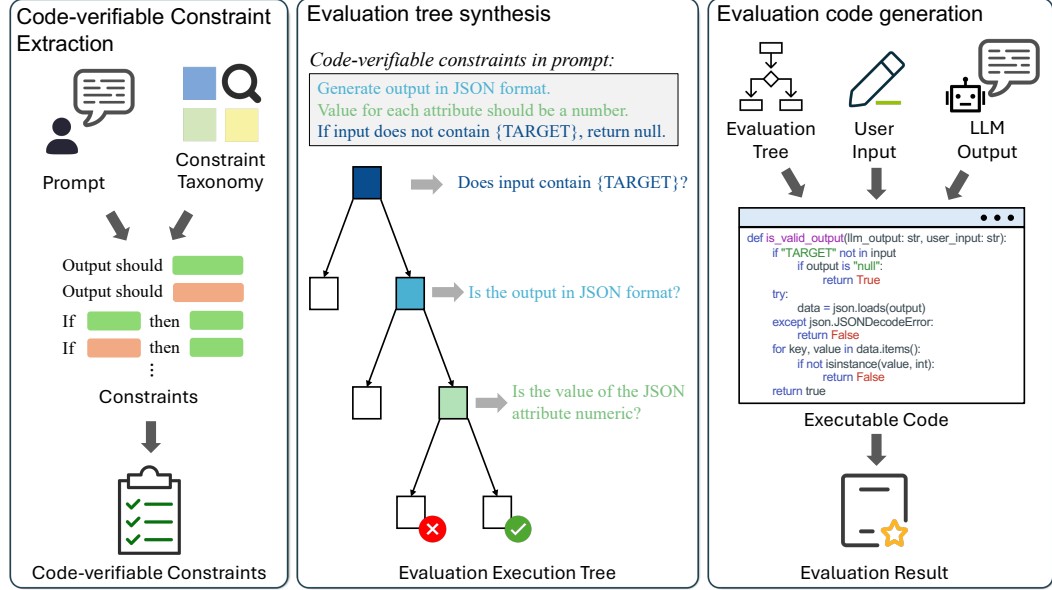

Figure 2: Pipeline overview

*generation*, which compiles the tree into an executable validator that returns deterministic pass/fail decisions together with human-readable reasons and provenance.

## 3.1 PROBLEM SETUP

Let $p$ be the prompt, $x$ the user input, and $y$ the model output. We represent each constraint $c$ as a pair $(g_c, r_c)$, where $g_c : \mathcal{X} \to \{0, 1\}$ is an optional *guard* over the input (the condition under which the constraint applies) and $r_c : \mathcal{X} \times \mathcal{Y} \to \{0, 1\}$ is the requirement over the output. Unconditional constraints use $g_c(x) \equiv 1$. The Boolean predicate realized by $c$ is

$$\phi_c(x, y) \;=\; \neg g_c(x) \;\vee\; r_c(x, y),$$

so the requirement $r_c$ is enforced exactly when the guard holds.

We focus on the code-verifiable families from Table 1: *specific format*, *numerical*, *lexical matching*, and *lexical exclusion*. Let $\Gamma(p)$ be the set of such constraints induced from $p$ (after extraction and filtering). The verifier returns *pass* iff all constraints hold:

$$\text{Verdict}(p, x, y) \;=\; \bigwedge_{c \in \Gamma(p)} \phi_c(x, y),$$

i.e., *pass* iff all constraints are satisfied.

## 3.2 CODE-VERIFIABLE CONSTRAINT EXTRACTION

> **Prompt 1: A Real-world Prompt example**
>
> ```
> Please analyze the following dialogue and evaluate it based on the criteria provided.
> Assign a score from 1 to 5 for each category. Scores should meaningfully distinguish
> quality levels. If the dialogue is less than 10 words, just return null.
> After your assessment, provide an overall score for the dialogue along with a concise
> summary. Please present your evaluation and comment into the following JSON format:
> {
>     "Relevance": _,
>     "Completeness": _,
>     "Correctness": _,
>     "Overall": {"score": _, "comment": _}
> }
> Here is the dialog: {dialog}
> ```

**Constraint extraction.** Given a prompt $p$, we first extract candidate constraints that specify requirements on the output. We prompt an LLM with our complete constraint taxonomy so that extracted spans can be mapped to categories with high recall and consistent definitions. For each detected constraint, the extractor returns (i) its *category* (we keep all families at this step), (ii) its *conditionality* (unconditional vs. triggered by an input condition), and (iii) the exact *source* span in the prompt.

On the running example Prompt 1, the extractor yields:

- *Specific format*: Output must be valid JSON with the keys Relevance, Completeness, Correctness, and Overall:{score, comment}.
- *Numerical*: Each score is an integer in $[1, 5]$.
- *Lexical (conditional)*: If the input dialogue has fewer than 10 words, the output must be "null".
- *Qualitative*: Scores should meaningfully distinguish quality levels.

**Code-verifiability filter.** We retain only constraints whose satisfaction can be decided deterministically by code (e.g., parsing, pattern matching, numeric tests over raw strings). Concretely, we first drop families outside the code-verifiable set (specific format, numerical, lexical matching/exclusion), which removes the qualitative constraint above, ensuring all remaining $r_c$ are code-verifiable. For conditional candidates with $g_c$, we test whether the trigger is itself verifiable from the input $x$ via surface-form checks (e.g., keyword containment, length thresholds). Non-verifiable guards are discarded. We prompt an LLM to judge this code-verifiability and return a Boolean verdict. In the running example, this filter keeps the conditional guard in "*return* null *if length* $< 10$". The output of Stage I is the set $\Gamma(p)$ of code-verifiable constraints that will be ordered and compiled in later stages.

### 3.3 EVALUATION TREE SYNTHESIS

**Benefit of tree.** Two observations motivate using a tree. (i) Prompts mix *global* constraints (whole-output format) with *local* ones (field/item rules). Local checks only make sense after the global structure is valid (e.g., verify values only once the output is valid JSON). (ii) Prompts may contain both *conditional* and *unconditional* rules. We check "conditional" constraints before "unconditional" ones. In practice, these are often not truly unconditional, but act as the *default* that applies exactly when no guard holds. For example: "*Always return JSON ... If the dialogue has $< 10$ words, output* null." Enforcing the JSON rule before evaluating the guard would incorrectly reject the intended null output. Therefore, we synthesize a guard-first, coarse-to-fine *evaluation tree* that specifies the correct order of constraint checks for subsequent step.

**Definition.** An evaluation tree $\mathcal{T}$ is a binary tree. Each internal node $u$ carries a small tuple of fields and has two outgoing edges: a *pass* branch (the check at $u$ holds) and a *fail* branch (otherwise). Leaves are labeled *result* $\in \{yes, no\}$. Every node exposes the following attributes:

- *conditional*: a bool indicating whether $u$ is an input guard that applies only to a subset of user inputs.
- *parent_ok*: a bool indicating whether, for a conditional node, the input guard is triggered; for an unconditional node, the prerequisite check for $u$ has been satisfied.
- *scope*: a string describes the granularity of the check (e.g., whole output, a specific key–value pair, an item in a list).
- *constraint_category*: the category the constraint belongs to (i.e., specific format, numerical, lexical matching, lexical exclusion).
- *constraint*: for a conditional node this is the input-side guard; for an unconditional node it is the code-verifiable output check.
- *source*: the exact span in the prompt that states the constraint, used for traceability and diagnostics that mirror the original wording.

Traversal of $\mathcal{T}$ proceeds as follows: If $u$ is *conditional*, evaluate only its guard on $x$ (e.g., "*the input has fewer than 10 words*") and take the corresponding branch; the required behavior (e.g., "*then return null*") is enforced in the pass child. If $u$ is *unconditional*, evaluate the output constraint on $y$. A pass continues along the pass branch to the next check, while a fail returns a *no* leaf.

**Constraint ordering policy.** We impose a canonical evaluation order for the tree:

1. **By granularity:** structure → field → value. Concretely, top-level format checks run before field presence and type checks, which in turn precede string and numeric value checks.

2. **Conditional-first within the same scope:** evaluate all conditional guards first. If an unconditional rule is intended as the *default* (i.e., it should apply only when no guard fires), place it on the *fail* branch of the relevant guard. Otherwise, place the unconditional check after both branches so that it is enforced regardless of the guard outcome.

3. **By prompt order:** any remaining checks follow their order of appearance in the prompt to preserve human expectations.

**Soundness.** If a traversal of $\mathcal{T}$ reaches a leaf labeled *yes*, then every compiled predicate on the path evaluated to true. Conversely, reaching *no* implies that at least one predicate failed, then we record the failing node's source span to support provenance.

**Synthesis.** Given the normalized constraint set from Stage I together with the node schema and ordering policy above, we prompt an LLM to synthesize the evaluation tree: the model instantiates nodes with (*conditional*, *parent_ok*, *scope*, *constraint_category*, *constraint*, *source*), arranges them into a guard-first, coarse-to-fine control flow, and returns a well-formed binary tree that the compiler consumes in the next stage. The evaluation tree provides an explicit control-flow specification that the code generator follows.

### 3.4 COMPILATION TO EXECUTABLE VALIDATORS

**Target interface and runtime.** We compile the specification into a single function

$$\texttt{is\_valid\_output}(y : \texttt{str}, x : \texttt{str}) \rightarrow (\texttt{bool}, \texttt{reason}, \texttt{provenance}),$$

executed in a sandbox. At runtime we pass *the LLM output* as $y$ and *the user input* as $x$ for the prompt under evaluation:

- **Input $y$ (LLM output).** The exact output string returned by the model.

- **Input $x$ (user input).** The user query or input text bound to the prompt; used only to evaluate input-side guards for *conditional* constraints.

- **Return bool.** True iff all code-verifiable constraints induced from the prompt are satisfied under the control flow prescribed by the evaluation tree; otherwise False.

- **Return reason.** A human-readable diagnostic describing the first failing check along the traversal.

- **Return provenance.** The exact prompt span (source) that defined the failed constraint, enabling traceability back to the original wording. Empty when the result is True.

**Inputs to code generation.** We synthesize code with an LLM by providing two inputs jointly: the original prompt $p$ and the evaluation tree from Stage II. The model is instructed to take ordering and scoping of checks from the tree while reading the complete context of $p$. This pairing aligns checks with the exact wording of $p$ and enforces them in the intended order.

**Generation rules.** To avoid hidden dependencies and keep execution reproducible, the generator must use *only* the Python standard library (e.g., json, re, typing). The generated code must not import third-party packages, perform file or network I/O, or invoke dynamic code execution. The output is a self-contained function with optional local helpers.

**Realizing the tree as control flow.** The synthesized function realizes the evaluation tree as directly executable code: it traverses nodes in tree order, evaluating input guards over $x$ to select the applicable branch (e.g., the conditional "*If the dialogue has fewer than 10 words*" becomes if word_count(x) < 10: and, on the pass branch, the code enforces the required behavior such as emitting the literal "null"); within the chosen branch it checks global structure on $y$ before any local content (e.g., "*The response must be a JSON object*" becomes try: parsed = json.loads(y) except: return (False, "not valid JSON", source)), then enforces field presence and types

(e.g., "*Keys Relevance, Completeness, Correctness must exist and be integers*" becomes `req = ["Relevance", "Completeness", "Correctness"]; ok = all(k in parsed for k in req) and all(isinstance(parsed[k], int) for k in req))`, and finally applies value-level predicates such as ranges, counts, and patterns (e.g., "*Each score is an integer in [1,5]*" becomes `all(1 <= parsed[k] <= 5 for k in req)`). Each predicate is evaluated at most once; the first failing check short-circuits with `False` together with a concise `reason` and the originating `source`, whereas successful checks continue along the pass edge until the traversal reaches a `yes` leaf and the function returns `True`.

**Runtime usage.** At runtime we apply the compiled validator to the pair $(y, x)$. The system returns `True` only if all constraints hold; otherwise it returns `False` with a concise reason and provenance.

## 4 EXPERIMENTS

### 4.1 EXPERIMENT SETUP

**Dataset.** We construct our own evaluation dataset because assessing constraint compliance requires prompt templates, instantiated inputs, model outputs, and ground-truth correctness labels with violation rationales. To our knowledge, no public dataset jointly provides all these components. We begin from the prompt corpus described in Section 2 (Data collection). The raw corpus consists of *prompt templates*. To obtain complete, runnable instances we supply concrete user inputs. Following the test–case generation pipeline of Sharma et al. (2025), we first filter for templates that contain only one user–input placeholder, which simplifies controlled input synthesis. For each retained template, we prompt an LLM to extract (i) *input rules* that characterize admissible user inputs for the template and (ii) *inverse rules* that deliberately violate these expectations to form a challenge set. Given the template and its rules, the LLM produces an initial pool of candidate inputs (three per template). A human evaluator then reviews the candidates for plausibility and refines them as needed, yielding a second–round set of inputs.

We instantiate the templates with these inputs and query target LLMs to obtain outputs. The annotators then judge whether each LLM output satisfies the prompt's code–verifiable constraints and, for failures, labels the *violated constraint* (the original constraint text from the prompt that was not met) and provides a minimal fix. For a given prompt there may be multiple violating output records and typically one fully compliant output record. The final dataset stores tuples *(Prompt, User input, LLM output, Correctness $\in \{True, False\}$, Violation)*. In total, the evaluation set contains 61 labeled records.

**Baselines.** We compare our verifier against an *LLM-as-a-judge* baseline for constraint–compliance evaluation: the target model is prompted to decide whether the output adheres to the prompt's code–verifiable constraints, and to justify its decision by returning the exact span of the original constraint text from the prompt.

**Implementation details.** We use GPT-4o to synthesize user inputs from templates. For evaluation, we instantiate our pipeline on three models for constraint extraction, evaluation–tree synthesis, and code generation: GPT-4o-2024-11-20 (Hurst et al., 2024), Deepseek-v3.1-non-thinking (DeepSeek-AI, 2024), and Claude-sonnet-4-20250514 (Anthropic, 2025). They cover both state-of-the-art open-source and closed-source models. Furthermore, Claude-sonnet-4 is specifically optimized for code generation, which aligns with our pipeline. For the LLM-as-a-judge baseline, we use the same model family for fairness. All models are accessed via official APIs with temperature set to 0. During code generation we allow up to three retries if the produced function fails to compile. All validators run in a sandbox and use only the Python standard library.

**Metrics.** We report **Constraint Compliance Accuracy** for the accuracy of the system's binary decision on whether the output satisfies all code-verifiable constraints, compared to the gold label. To assess explanation quality, we compute **Violation Rationale**, the BLEU similarity between the system's failure reason and the human–annotated rationale for the violated constraint. If the system's compliance decision disagrees with the gold label, we set Violation Rationale to 0. Otherwise, we compute BLEU on the paired rationales and average over the dataset. Together, these metrics capture both correct compliance decisions and the fidelity of the accompanying justification.

| Model | Constraint Compliance Accuracy | | Violation Rationale | |
|---|---|---|---|---|
| | LLM-as-a-judge | Our method | LLM-as-a-judge | Our method |
| GPT-4o | 62.3 | **86.9 (+39.5%)** | 0.3174 | **0.6137 (+93.4%)** |
| Deepseek-v3.1 | 68.9 | **82.0 (+19.0%)** | 0.4361 | **0.4937 (+13.2%)** |
| Claude-sonnet-4 | 80.3 | **91.8 (+14.3%)** | 0.5221 | **0.6042 (+15.7%)** |

Table 2: Main results.

## 4.2 Experimental results and analysis

**Main results.** As shown in Table 2, our method consistently outperforms a plain *LLM-as-a-judge* baseline across all three models (GPT-4o, Deepseek-v3.1, Claude-sonnet-4) on both *Constraint Compliance Accuracy* and *Violation Rationale*. Measured as a macro–average of per–model relative gains, Constraint Compliance Accuracy improves by **+24.3%** overall (range from +14.3% to +39.5%), and Violation Rationale improves by **+40.8%** (range from +13.2% to +93.4%). The largest gains appear with GPT-4o: +39.5% on Constraint Compliance Accuracy (62.3→86.9) and +93.4% on Violation Rationale (0.3174→0.6137), indicating that compiled, deterministic checks recover a substantial number of hard–constraint violations that LLM-as-a-judge misjudges or overlooks. Gains remain positive for Deepseek-v3.1 (+19.0% Constraint Compliance Accuracy, +13.2% Violation Rationale) and Claude-sonnet-4 (+14.3%, +15.7%), demonstrating that the approach generalizes across model families. Across the board, our method yields consistently higher or comparable performance compared with each model's strongest LLM-as-a-judge baseline on both metrics.

These improvements are most pronounced on two patterns: (i) *Hybrid granularity*: prompts that mix global structure with field-level or item-level checks. In such cases, *LLM-as-a-judge* often mis-scopes constraints. For example, consider the requirements "*the output must be a list*" and "*the second element must itself be a list of exactly five items*", given the output `['True', ['CRISPR', 'genome', 'editing', 'bioinformatics', 'databases'], '100']`, the baseline judged: "*The output contains 6 words in the list for Part B instead of the required 5*", erroneously counting tokens from the outer list rather than the nested list. Our method correctly parses the structure and checks the length on the second element only. (ii) *Closed-format exclusivity*: prompts that impose both exclusivity (no out-of-format content) and completeness (all specified elements present) at the same scope. For the constraint "*Only respond with the format below using curly brackets to encapsulate the variables within a JSON dictionary object and no other text*", the baseline occasionally accepts outputs prefixed with extra markers (e.g., a leading ```json tag) despite the "*no other text*" rule, whereas our compiled checks flag any extraneous content outside the JSON object. By compiling prompt constraints into a guard–first, coarse–to–fine control flow with explicit scopes and executing code-based checks, our method reduces such misses and produces diagnostics that align closely with gold violation rationales. Overall, making constraint verification explicit and executable proves model–agnostic, complements LLM-as-a-judge judging, and yields robust improvements on real prompts.

**Ablation study** Table 3 isolates the contribution of the evaluation tree by comparing code generated from a flat list of constraints ("Without tree") to tree–guided code generation ("With tree"). Across all three models (GPT-4o, Deepseek-v3.1, Claude-sonnet-4), adding the tree improves both *Constraint Compliance Accuracy* and *Violation Rationale*. Accuracy rises by **+3.9%** (GPT-4o), **+6.5%** (Deepseek-v3.1), and **+14.3%** (Claude-sonnet-4), for an average gain of **+8.2%**. Explanation quality (*Violation Rationale*) also increases consistently by **+6.1%**, **+6.6%**, and **+5.6%** respectively (avg. **+6.1%**). The improvement is most pronounced for Claude-sonnet-4, whose strong software–engineering capabilities enable it to precisely recognize and leverage the rich auxiliary context provided during code generation (e.g., ordering hints, scope annotations, and schema comments), yielding higher–quality validators.

Qualitatively, the tree reduces ordering and scoping errors that arise when checks are emitted from a flat set of constraints: input guards are evaluated first, and global structure is verified before field–level presence/typing and value–level limits. This guard–first, coarse–to–fine control flow prevents false rejections of legitimate corner cases (e.g., outputs that should be the literal "null" and avoids overlooking downstream violations (e.g., malformed JSON or range overruns). The explicit branching also yields crisper, more reproducible rationales by tying each failure to the correct path and constraint, which explains the consistent gains in both metrics.

| Method | Constraint Compliance Accuracy | | Violation Rationale | |
|---|---|---|---|---|
| | Without tree | With tree | Without tree | With tree |
| GPT–4o | 83.6 | **86.9 (+3.9%)** | 0.5786 | **0.6137 (+6.1%)** |
| Deepseek–v3.1 | 77.0 | **82.0 (+6.5%)** | 0.4630 | **0.4937 (+6.6%)** |
| Claude–sonnet–4 | 80.3 | **91.8 (+14.3%)** | 0.5723 | **0.6042 (+5.6%)** |

Table 3: Ablation on the evaluation tree. "Without tree" compiles validators directly from extracted constraints; "With Tree" additionally uses the evaluation tree to order and scope checks.

## 5 RELATED WORK

**Controlled text generation.** A large body of research steers language models toward user–specified constraints, spanning both *decoding–time* control and *training–time* control. Decoding–time methods inject hard lexical requirements into search (Hokamp & Liu, 2017; Post & Vilar, 2018) or bias token probabilities with auxiliary discriminators and predictors (Dathathri et al., 2020; Khanov et al., 2024; Li et al., 2025; Yang & Klein, 2021; Lu et al., 2021). Training–time approaches encode control signals directly in the model, either via control codes or preference–based fine–tuning (Keskar et al., 2019; Krause et al., 2020; Ouyang et al., 2022). While these methods substantially *reduce* violations, they rarely *guarantee* compliance in real prompts that mix global formats with field–level or item–level rules and conditional exceptions. Our work is complementary: instead of steering generation, we *verify after the fact* by inducing code-verifiable constraints from the prompt, organizing them with an explicit control flow, and executing deterministic checks.

**Evaluation of LLM outputs.** LLM–as–a–judge has become a scalable alternative to costly, slow human evaluation across many generation tasks (Gu et al., 2024). It is widely used to rate output quality (e.g., semantic correctness, coherence, helpfulness, safety) (Liu et al., 2023; Kim et al., 2024; Zhu et al., 2025) and to approximate human preferences for training and benchmarking (Dong et al., 2023; Yuan et al., 2023; Dubois et al., 2023). However, recent work highlights unfairness and sensitivity to prompt wording and presentation order (Wang et al., 2024; Zheng et al., 2023). In practice, such judges excel at soft qualities but often under–penalize hard constraints (e.g., malformed JSON, missing required fields, or length overruns), and their rationales can vary across prompts and models. By contrast, rule–based scripts offer deterministic, reproducible checks for hard constraints (Jie et al., 2024; Carlsson et al., 2022; Wang et al., 2021), yet they are typically ad hoc and coarse–grained, often operating at a single granularity that treats the output as a monolithic string rather than aligning checks with constraint granularity or handling conditional guards. As a result, such scripts become brittle and hard to maintain when constraints mix global structure, local fields, and input–conditioned behaviors. We bridge this gap by compiling prompt–induced, code–verifiable constraints into executable validators with a canonical coarse–to–fine order, yielding reproducible pass/fail decisions and grounded rationales that complement the existing LLM output evaluation methods while substantially improving coverage of hard constraints.

## 6 CONCLUSION

We introduce *ConstrainPrompt*, a verification pipeline for code-verifiable, semantics-agnostic constraints in LLM outputs. Our taxonomy study of 1,232 real prompts shows that hard constraints are prevalent and often layered-global structure commonly co-occurs with field-level and item-level rules and input-conditioned guards. ConstrainPrompt induces code-verifiable constraints from natural-language prompts, synthesizes a logical evaluation tree to fix ordering and scope, and compiles the result into executable validators that return deterministic pass/fail decisions with grounded rationales and prompt-span provenance. On real prompts paired with model outputs, our method consistently outperforms an LLM-as-a-judge baseline across three model families, improving Constraint Compliance Accuracy by 24.3% and Violation Rationale by 40.8%. An ablation study confirms the evaluation tree is pivotal for both accuracy and explanation quality. Overall, these findings demonstrate that making constraint verification explicit and executable is an effective, model-agnostic method for reliable constraint compliance in real-world LLM applications.

# 7 REPRODUCIBILITY STATEMENT

We provide an anonymized repository containing the full pipeline and scripts to reproduce all reported results: `https://anonymous.4open.science/r/ConstrainPrompt-C278/`. The repository includes implementations for constraint extraction, evaluation tree synthesis, and evaluation code generation. All core prompts used for LLM interactions are included verbatim in the Appendix. The exact model families and versions used in each experiment are specified in the Experiment Setup section, and the released scripts invoke those versions by default.

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

## A    LLM Contribution Statement

In accordance with the ICLR policy, we disclose that large language models (e.g., OpenAI ChatGPT and Google Gemini) were used solely as writing aids for: (i) improving grammar, fluency, and stylistic consistency; (ii) rephrasing sentences for clarity; and (iii) polishing LaTeX table code. LLMs did not generate research ideas, methods, experiments, analyses, figures/tables (beyond LaTeX formatting templates), or claims, and no passages were adopted verbatim without manual review and editing by the authors. All technical content, datasets, code, and results were produced and verified by the authors, who take full responsibility for the paper's contents.

## B    Constraint taxonomy examples

| Category | Subcategory | Examples |
|---|---|---|
| Structural | Specific format | - Return output in json format: "question_1": "answer_1", "question_2": "answer_2", ..., "question_n": "answer_n". 
 - Output a list with specific entries: Entry 1 as a string response to Part A, Entry 2 as a list of 5 words from Part B, and Entry 3 as an integer response to Part C converted to a string. |
| Quantitative | Numerical | - Keep your wording simple and short (less than 15 words). 
 - Provide your evaluation only as a consistency score where the consistency score is an integer value between 0 and 5, with 5 indicating the highest level of consistency. |
| Lexical | Lexical matching | - Respond to each question from the provided "questions", using either "Yes", "No", or "Unknown", based on the given context. 
 - Give me only the sql query statement, starting with "SELECT * FROM Verses WHERE" without any extra explanation or comment. |
| | Lexical exclusion | - Please write a quiz question for the word "horse" using single sentence without mentioning the word itself. 
 - Please do not fill in "unknown", but make an educated guess based on the available information and fill in the specific content. |
| Semantic | Semantic inclusion | - Identify any controversial or heavily debated points in the video. 
 - Ensure to extract the key insights, theories, steps, revelations, opinions, etc discussed in the video. |
| | Semantic exclusion | - You should not describe items in the image include people's faces, hands, text or animals, device screens or anything that could contain text. 
 - Remove any of the following sentences: Sentences that refer to grammar, spelling or punctuation. Sentences that say the response is unclear or not concise enough. Sentences that give away the correct answer explicitly. |
| | Qualitative | - The refined blog post text should be easy to read, and engaging. 
 - The analyzer should address all user's queries about the TV show in a concise, friendly, conversational style. |

Table B1: Constraint taxonomy examples.

## C    Prompts

### C.1    Constraint extraction

```
Constraint extraction prompt

You are a precision prompt constraint analyzer.

Given a prompt template, extract **all constraints** that specify what the model output
    must or must not do. For each constraint:
- Determine whether it's **unconditional** or **conditional**:
  - **unconditional**: This constraint applies universally to all outputs, regardless of
    the input content.
```

```
 - **conditional**: The prompt specifies a trigger condition that determines whether the
   constraint applies, and the trigger is either:
   - Expressed using clear indicators like "if...", "when...", "only if...", "in case...",
     or
   - Explicitly tied to detectable input features (e.g., contains specific keywords,
   matches a language code, exceeds a given length).
- Assign one of the following categories:
  1. Output - Specific format constraint: The output must conform to a specific file or
     data format (e.g., JSON, Markdown, HTML, key value pairs, defined data structures,
     source code in a specific language).
  2. Output - Numerical constraint: Restrictions on output length, counts, or numerical
     ranges (e.g., character/word/token/sentence/paragraph counts, score values).
  3. Output - Lexical matching constraint: The output must contain, match, or adhere to a
     specific string pattern (e.g., selection from a predefined list, exact string match,
     lowercase requirement).
  4. Output - Lexical exclusion constraint: Certain words, phrases, string or character
     patterns are explicitly prohibited in the output.
  5. Output - Semantic inclusion constraint: The output must semantically include certain
     concepts, entities, or topics. (not verifiable by code)
  6. Output - Semantic exclusion constraint: The output must not semantically mention
     certain concepts, entities, or topics. (not verifiable by code)
  7. Output - Qualitative constraint: The output must exhibit specific non-quantitative
     qualities or styles (e.g., concise, academic tone, persuasive, language). (not
     verifiable by code)
  8. Others
- For each constraint, extract the **exact sentence** from the prompt that expresses it as
    'source'
- Give a short justification for the category.

Return all constraints in a structured list using the function tool.
```

## C.2 EVALUATION TREE SYNTHESIS

**Evaluation tree synthesis prompt**

```
You are a constraint-checking logic tree generator.

Your task is to construct a single decision tree for validating model output based on a
    list of constraints.

Each node must include the following fields:
- conditional: true or false
- parent_ok: true or false
- constraint_category: one of the following: 'Output - Specific format constraint', '
    Output - Numerical constraint', 'Output - Lexical matching constraint', 'Output -
    Lexical exclusion constraint', or 'result' (used in leaf nodes)
- constraint: a concise human-readable description of what is being checked. **Also
    clearly indicate whether this applies to the entire output or to a specific field/
    section.** For example:
  - "output must be valid JSON object"
  - "output['queries'] must be a list of 1 to 5 unique strings"
- source: exactly copy the `source` field from the corresponding constraint object
    provided in the corresponding constraint in the input constraint list.
- scope: A description of **which part of the output** this constraint applies to.
  - Use "entire output" if the constraint applies to the whole response (e.g., length,
    general formatting, string content).
  - Use specific references (e.g., "JSON field 'questions'", "markdown header", "list
    elements", "first sentence") when the constraint targets a **subsection or component**
     of a structured output.
- children: a list of exactly two children unless this is a leaf node; children must
    describe what happens when the constraint **is met** and **is not met**
```

```
Rules:
1. The tree must evaluate constraints **in this order**:
   - First: all **conditional** constraints
   - Then: all **unconditional** constraints
   - Within each group: order by granularity - format - type/field - value
   - This reflects a macro-to-micro validation order: check overall structure first (e.g.,
     JSON), then expected output type, then more detailed content or length.

2. Each conditional constraint must have two child branches:
   - If **condition is met** ('parent_ok=True'): its expected output behavior must be
     explicitly checked as a child node. Only apply the behavior required by that
     conditional constraint
   - If **condition is not met** ('parent_ok=False'): evaluate all unconditional
     constraints in order

3. For **every constraint node**, generate exactly **two children**:
   - One where 'parent_ok = true' (constraint is satisfied)
   - One where 'parent_ok = false' (constraint is not satisfied)

4. All **leaf nodes** must be of the form:
   - 'conditional': false
   - 'parent_ok': true or false
   - `'onstraint_category': 'result'
   - 'constraint': 'yes' or 'no'
   - 'source': `None`
   - 'scope': `None`
   - 'children': empty list

5. For conditional constraints, only evaluate the condition at the current node.
   - If the condition is met (parent_ok = true), generate a child node to check the
     required action or constraint specified by the condition.
   - If the condition is not met (parent_ok = false), proceed to check other related
     constraints.
```

## C.3 EVALUATION CODE GENERATION

**Evaluation code generation prompt**

```
You are a Python code generation agent specialized in logical validation.

Your task is to generate a Python function `is_valid_output(output: str, input_text: str)
    -> Tuple[bool, Optional[str], Optional[str]]` that checks whether the model output
    satisfies a set of constraints, which are organized as a decision tree.

The decision tree follows this format:
- Each node contains:
  - 'conditional': whether this is a conditional constraint (True/False)
  - 'parent_ok': whether the condition of its parent node was satisfied (True/False)
  - 'constraint_category': the category of the constraint or 'result' if leaf node
  - 'constraint': a human-readable string that describes the check
  - 'source': the source constraint (the original sentence in prompt) that represents the
    check in evaluation tree
  - 'scope': the part of the output this constraint applies to
  - 'children': list of child nodes

### Rules:
1. Before any validation, normalize the `output` string to prevent false negatives from
    insignificant whitespace:
   - Strip leading and trailing blank lines
   - Collapse multiple consecutive blank lines into a single blank line
```

```
2. Traverse the tree starting from the root node. At each node:
   - If `constraint_category == 'result'`, return `(True, None, None)` if `constraint == '
     yes'`, else return `(False, <short reason>, <violation>)`, where <reason> is a
     concise diagnostic describing what failed and <violation> is the constraint's `source
     ` value in evaluation tree (the original prompt constraint that was violated).
   - If `conditional == True`, evaluate the *condition part only* of the constraint at
     this level
      - If the condition is **not directly verifiable in code**, return False by default
      - If the condition is verifiable, use an `if` to decide which branch of `children`
     to check
   - If `conditional == False` and `parent_ok == True`, validate the constraint against
     the output
      - If it passes, recurse to `children[0]`; else recurse to `children[1]`
      - If `parent_ok == False`, directly recurse to `children[1]`

3. The `output` is always the raw string returned by a language model. Any structural
   checks (e.g., JSON parsing) or type checks (e.g., numeric value) must first convert or
    parse this string appropriately.

4. The generated code must handle malformed output robustly (e.g., invalid JSON)

5. You may define helper functions for checking common patterns (e.g., word count, JSON
   keys, exact match)

6. The constraint string should be used as a comment to make clear what each check is
   doing.

7. Only use standard Python libraries (no external dependencies).

Output the complete Python function only. Do not include explanation or comments outside
   the code.
```

# D CASE STUDY

## D.1 PROMPT, USER INPUT, LLM OUTPUT

**Prompt**

```
I will provide you with a prompt to a function below these instructions. You will output
    exactly as follows, with the list as well. Text encased in <like this> will be
    replaced by your response, and text encased in (like this) is just a description for
    the response that you do not need to type up:
  (a) <Boolean> (Is it bioinformatics related?)
  (b) <words> (Give a list of 5 keywords of why it is bioinformatics related)
  (c) <integer> (Your confidence from 0 to 100 that your response in A is accurate, so for
      example, if you believe strongly that it is not bioinformatics related, you should
      also rate a high confidence level)
  The code must explicitly reference some bioinformatics methodology, terminology, or
    process. For example, an AVL Tree would not be a valid bioinformatics function, while
    a FASTQ processor would. The keywords are defined as important words that allowed you
    to make the determination that the function is bioinformatics related. The confidence
    should be your estimate of how confident you are of your responses.

  Make sure that in your response is explicitly as follows in the directions. Part A
    should only be one word and a boolean, either True or False. Part B should only be 5
    words, no additional information, Part C should only be a single integer, from 0 to
    100, it is a measure of your confidence in your response to Part A.

  After selecting keywords, please reverify that the words you used to make the decision
    for Part A is actually bioinformatics related.
```

```
Again, as clarification, I will be providing the function.

The responses should be formatted as a list:
Entry 1: The response to part A converted into a string
Entry 2: A list of 5 words which are strings from the response to Part B
Entry 3: The integer response to part C converted to a string
Therefore, your output should follow this guideline. This will be your only output,
    there should be no additional outputs beyond the one highlighted in this prompt.

Prompt begins here:
{input_prompt}

Prompt ends here.
Give the output to the above code encased in "Prompt begins here:" and "Prompt ends here
    ." Your keyword search should only encompass the words in the prompt, and ensure that
    keywords are related to bioinformatics, not statistics.
```

**User input**

```
The program processes CRISPR data to predict off-target effects in genome editing
    applications, leveraging bioinformatics databases and algorithms.
```

**Output**

```
['True', ['CRISPR', 'genome', 'editing', 'bioinformatics'], '100']
```

## D.2 EVALUATION TREE

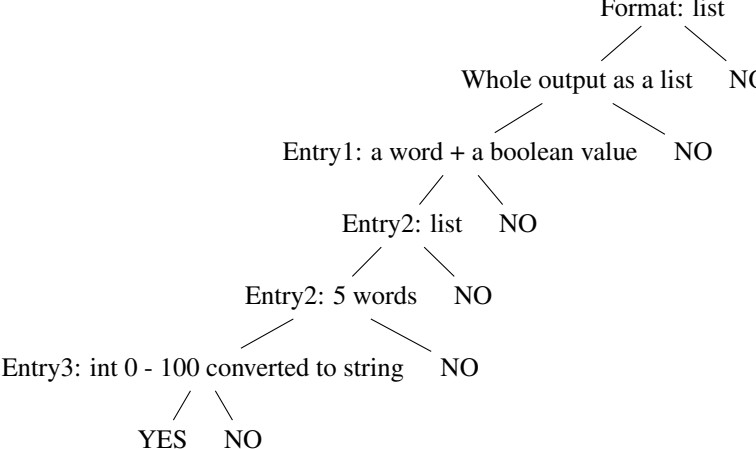

Figure 3: Evaluation tree for the example.

## D.3 GENERATED CODE

**Generated code**

```python
from typing import Tuple, Optional
import re

# Helper function to check if a string is a valid integer within a range
```

```python
def is_valid_integer(s: str, min_value: int, max_value: int) -> bool:
    try:
        value = int(s)
        return min_value <= value <= max_value
    except ValueError:
        return False

# Main function to validate the output

def is_valid_output(output: str, input_text: str) -> Tuple[bool, Optional[
    str], Optional[str]]:
    # Normalize the output
    output = re.sub(r'\n+', '\n', output.strip())

    # Check if the output is a list of three entries
    try:
        entries = eval(output)
        if not isinstance(entries, list) or len(entries) != 3:
            return False, "Output is not a list of three entries", "The
                responses should be formatted as a list: Entry 1: The
                response to part A converted into a string Entry 2: A list
                of 5 words which are strings from the response to Part B
                Entry 3: The integer response to part C converted to a
                string"
    except:
        return False, "Output is not a valid list", "The responses should
            be formatted as a list: Entry 1: The response to part A
            converted into a string Entry 2: A list of 5 words which are
            strings from the response to Part B Entry 3: The integer
            response to part C converted to a string"

    # Check Entry 1: Part A should only be one word and a boolean, either
        True or False.
    if not isinstance(entries[0], str) or entries[0] not in ["True", "False
        "]:
        return False, "Entry 1 is not a valid boolean string", "Part A
            should only be one word and a boolean, either True or False."

    # Check Entry 2: Part B should only be 5 words, no additional
        information.
    if not isinstance(entries[1], list) or len(entries[1]) != 5 or not all(
        isinstance(word, str) for word in entries[1]):
        return False, "Entry 2 is not a list of 5 words", "Part B should
            only be 5 words, no additional information."

    # Check Entry 3: Part C should only be a single integer, from 0 to 100.
    if not isinstance(entries[2], str) or not is_valid_integer(entries[2],
        0, 100):
        return False, "Entry 3 is not a valid integer string", "Part C
            should only be a single integer, from 0 to 100."

    # If all checks pass, return True
    return True, None, None
```

