# OpenReview forum: "ConstrainPrompt: Code-Based Assurance of Prompt-Defined Constraints"
_ICLR.cc/2026/Conference — Submitted to ICLR 2026_

### Official Review · Reviewer_oYjk · 2025-10-31

**Soundness:** 4
**Presentation:** 3
**Contribution:** 2
**Rating:** 6
**Confidence:** 3

**Summary:**

This paper focuses on post-generation validation rules for LLM. It proposes ConstrainPrompt, a pipeline that automatically extracts constraints for the user input, deduces a tree-like structure for validation, and adopts LLM to generate the final code-based validation test. The paper also builds a new benchmark for evaluating how accurate ConstrainPrompt can validate outputs generated from real-world prompts. In the evaluation, the paper shows that ConstrainPrompt outperforms the vanilla LLM-as-a-judge. The ablation study also shows the effectiveness of the judgement tree.

**Strengths:**

- The paper focuses on a novel problem that is prevalent for people that adopt LLM into their workflows.
- The paper is presented well, with nice flow and comprehensive quantitative evaluation.

**Weaknesses:**

- The importance of the problem is somehow questionable. Normally people derive prompts and then handwrite validation rules as a one-time effort. ConstrainPrompt only handles mostly syntactic checking, which already does not take much effort.
- The evaluation is missing some details. For example, for how many times are the evaluation run? Also it is unclear which model generated the outputs for the benchmark.

**Questions:**

- Why is the problem important? How is ConstrainPrompt much better than human-written validation, which is only a one-time effort per prompt?
- What is the fluctuation of the result? How does the fluctuation affect the result of the ablation study?
- Which model is used to generate the outputs for the benchmark?

---

> ### Author Response · Authors · 2025-11-20
>
> Dear Reviewer oYjk,
>
> Thank you very much for your thoughtful and constructive feedback. We sincerely appreciate the opportunity to clarify the motivation and importance of our problem setting.
>
> **Regarding the importance of the problem:** we fully acknowledge the intuition that practitioners may write validation rules manually as a one-time effort. However, this assumption rarely holds in real LLM-based application development. In practical production workflows, prompts are updated frequently as part of routine maintenance. Prompt engineers and DevOps teams continuously revise instructions to improve reliability, prevent regressions, or adapt to new product requirements. Even seemingly minor modifications to a prompt may alter the expected output structure or semantics.
>
> Once prompts evolve, manually validating the new outputs becomes extremely laborious. Developers must repeatedly inspect model outputs line by line, and even experienced engineers often overlook subtle violations such as off-by-one numeric mistakes, boundary-condition failures, incorrect length constraints, or small structural omissions. These issues may appear minor in natural text but can be critical when outputs feed into downstream programs or operational pipelines. The cumulative risk and effort grow quickly as prompts change, making manual inspection inefficient and unreliable.
>
> In addition, writing rule-based validators is not always feasible in practice. Many prompt engineers lack the ability to implement robust and generalizable validation scripts, especially when prompts cover diverse formats or complex multi-field structures. Even for teams with such expertise, maintaining a separate custom validator for every evolving prompt introduces substantial engineering overhead and does not scale well across larger systems or fast-moving development cycles.
>
> ConstrainPrompt is designed specifically to address these challenges. By automatically extracting code-verifiable constraints, synthesizing validation logic, and applying consistent checking across different prompt versions, our method reduces the human burden and significantly lowers the likelihood of missing subtle but important violations. This automated evaluation pipeline provides a more reliable and maintainable alternative to repeated manual inspection or ad-hoc hand-written validators, especially in environments where prompts evolve continuously and correctness guarantees are essential.
>
> **Regarding result fluctuation and experiment stability:** In all experiments, we set the temperature of every model to the lowest possible value to minimize randomness during generation. In our main experiments, we observed that the number of correctly or incorrectly judged cases fluctuates very little across repeated runs. The variation in raw case counts is within two for Constraint Compliance Accuracy. Based on these observations and in order to ensure both stability and fairness, we conducted three runs for each model and reported the best score among the three for both the baseline and our method.
>
> **Regarding the model used for generating benchmark outputs:** For constructing the benchmark outputs, we first used GPT-4o to synthesize the input examples for each prompt template. Then, to obtain diverse and potentially imperfect outputs including those exhibiting various kinds of instruction-following failures, we generated the filled-prompt outputs using both GPT-4o and Gemini 2.5 Flash. These outputs were subsequently analyzed manually to build the final evaluation set. This process ensures the dataset contains a broad range of realistic model behaviors, rather than being biased toward a single model’s generation pattern.
>
> We truly appreciate your insightful comments and thank you again for taking the time to review our submission.

---

### Official Review · Reviewer_ga62 · 2025-10-31

**Soundness:** 1
**Presentation:** 3
**Contribution:** 2
**Rating:** 2
**Confidence:** 3

**Summary:**

The paper presents ConstrainPrompt, a verification pipeline that separates constraint compliance from semantic quality to ensure LLM outputs satisfy hard control constraints. ConstrainPrompt identifies hard control constraints (e.g., format, lexical inclusion, length limits) from prompts, organizes them into a logical evaluation tree that enforces a global-to-local validation order, and compiles this tree into executable Python validators. Experiments show that ConstrainPrompt outperforms the LLM-as-a-judge baseline across three models in both constraint compliance accuracy and violation rationale quality.

**Strengths:**

1. The method solves a meaningful task. It ensures that the generated results satisfy the prompt defined hard control constraints, for example, format, lexical inclusion, and length limits.

2. The guard-first, coarse-to-fine ordering enforces logical precedence between global and local constraints, improving robustness and interpretability.

3. The method demonstrates significant performance in terms of compliance accuracy and violation rationale.

4. The manuscript is easy to follow, and the method is clearly described and defined.

**Weaknesses:**

1. LLM-as-a-judge is a weak baseline. There are many works related to agentic workflow that address the same issue. Also, constrained decoding is a typical way but the paper didn’t compare them them.

2. Current instruction-tuned models can already align well with complex output requirements (e.g., GPT-4.1 performs much better than GPT-4o on instruction following [1]). In addition, instruction-tuned models can align with any kind of output requirement. At the same time, constrainprompt did not mention how to generalize to those output requirement beside promptset (line 93). In other words, the constraints are not generalizable.

3. The evaluation relies on a single benchmark, 61 records only (Line 355), which could not represent the real-world diversity. Further experiments on related benchmarks are necessary.

4. The manuscript mentioned in line 55, one method to check the output constraint is a rule-based script. But no related baselines in the experiment. Also, the experimented models are limited to only 3 powerful LLMs (not enough). It is not clear whether the pipeline can benefit smaller LLMs.

5. Extraction, tree synthesis, and code generation all rely on LLMs (Sec. 3); there is a risk that they introduce bias or errors from the LLM itself. Related failure cases are not discussed.

6. The generated validators could add much more computational overhead, while efficiency analysis is not discussed.

[1] https://openai.com/index/gpt-4-1/

**Questions:**

As current instruction-tuned models increasingly follow prompts accurately, how does ConstrainPrompt + base model compare against instruction-tuned models alone in terms of constraint compliance?

For the constraints category beyond PromptSet, how does ConstrainPrompt generalize?

---

> ### Author Response · Authors · 2025-11-20
>
> Dear Reviewer ga62:
>
> Thank you very much for your thorough and valuable comments. We sincerely appreciate the opportunity to clarify and strengthen our work.
>
>
> ### Regarding baseline selection
>
> **Agentic workflow and constrained decoding:** We fully understand the concern about including additional baselines. However, to the best of our knowledge, there is currently no mature agentic workflow specifically designed for prompt-constraint compliance evaluation, which is the focus of our work. Most agentic systems are designed for multi-step problem solving, planning, or tool use, rather than fine-grained constraint verification. Similarly, constrained decoding methods primarily target the generation phase (i.e., ensuring outputs adhere to constraints during decoding), while our method focuses on the evaluation phase (i.e., accurately verifying constraint compliance after generation). These approaches are complementary rather than comparable. In fact, constrained decoding can be combined with our method to create a more reliable end-to-end controlled generation pipeline. However, they do not provide a direct alternative to the evaluation mechanism we propose.
>
> **Rule-based validators:** We agree rule-based baselines is relevant, but such systems require heavy manual engineering and are inherently task-specific. As an example, prior work such as [1] measures keyword-inclusion coverage for a single dataset and corresponds to only one fixed prompt template in our context. Our goal is to support arbitrary and diverse real-world prompts, making manually crafted rule-based systems non-scalable. For this reason, while rule-based approaches may be useful in narrow domains, they do not meaningfully generalize to the broad, heterogeneous prompt formats targeted by our method. That’s why we do not take the rule-based method as our baseline.
>
>
> ### Regarding model selection
>
> **Comparison with instruction-tuned models:** To investigate whether our method can compensate for a base model’s lack of instruction following compared to an instruction-tuned model, we add a new experiment comparing *Qwen3 235B Base + our method* against *Qwen3 235B Instruction-tuned*. Qwen3 235B (instruction-tuned) achieves **77.3** in Constraint Compliance Accuracy and **0.5307** in Violation Rationale, whereas Qwen3 235B (base) + our method reaches **75.8** and **0.4989**, representing a relative decrease of **1.9%** and **6.0%**. This shows that although our approach does not elevate the base model to consistently outperform the instruction-tuned version, it does narrow the gap. Our method relies on the model’s instruction-following ability for correctly extracting and analyzing constraints from the prompt, which is a prerequisite for generating accurate validation code. Therefore, the performance of ConstrainPrompt is partly bounded by the model’s inherent instruction-following capability.
>
> **Evaluation on weaker models:** We also extend our experiments to *Qwen3 14B* and *Gemma3 12B*. Our method improves both metrics on Gemma3 12B (**+40%** and **+8.9%**), but degrades performance on Qwen3 14B (**−4.8%** and **−10.5%**). After analyzing the results, we find that the difference in performance between the two models primarily stems from their disparities in code-generation ability. Qwen3 14B exhibits a **15.2%** failure rate in code generation (including compilation and runtime errors), whereas Gemma3 12B has a much lower failure rate of only **6.1%**. These results show that our method can enhance the behavior of weaker models by leveraging their code-understanding ability, but models with insufficient code-generation capability can become a bottleneck.
>
> ### Regarding dataset size and generalizability
>
> **Dataset scale:** We fully understand the concern about the relatively small dataset. Unfortunately, there is currently no large scale public benchmark specifically designed for real-world prompt-constraint compliance. As such, we must manually inspect every prompt template and label each generated output, which significantly limits dataset size. Expanding the dataset is an important direction for future work.
>
> **Generalizability:** Our dataset is constructed from PromptSet, which collects prompt templates from diverse real-world GitHub repositories including different task types, domains, difficulty levels, and usage contexts (from demo projects to production systems). We therefore believe the collected prompts already cover a broad spectrum of realistic scenarios and provide reasonable generalizability.
>
> [1] Carlsson, Fredrik, et al. "Fine-grained controllable text generation using non-residual prompting." Proceedings of the 60th Annual Meeting of the Association for Computational Linguistics (Volume 1: Long Papers). 2022.

---

> > ### Author Response · Authors · 2025-11-20
> >
> > Dear Reviewer ga62:
> >
> > In this follow-up reply, we address the rest of the reviewer’s comments in detail.
> >
> > ### Regarding potential bias and errors in LLM-based steps
> >
> > Regarding potential LLM errors in the code-verifiable constraint extraction and tree synthesis stages, we fully acknowledge that inaccuracies introduced by these LLM-driven components can propagate through the pipeline. To better understand their impact, we conducted a manual inspection by randomly sampling 20% of both successful and failed cases and carefully evaluating each step. Our analysis shows that, in successful cases, Step 1 (prompt → code-verifiable constraints) achieves **90.9%** accuracy, and Step 2 (constraints → evaluation tree) achieves **95.5%**. In contrast, for failed cases, the accuracy drops to **27.3%** in Step 1 and **81.8%** in Step 2. These results indicate that most failure cases originate from misclassification during the code-verifiability filtering step, which highlights an important direction for future refinement of our pipeline.
> >
> > ### Regarding computational overhead
> >
> > Thank you for raising this important point. ConstrainPrompt is indeed primarily designed for offline evaluation, not real-time inference. There are two key applications where the overhead is acceptable and necessary:
> >
> > **LLM application testing during development and updates:** In real-world LLM-based applications, prompt templates evolve with model updates or feature changes. Even small modifications can alter output behavior significantly. Developers and operation engineers must therefore repeatedly test a sample set of prompts to ensure correctness. Manually inspecting constraint compliance is extremely time-consuming, while our method provides a systematic, repeatable evaluation process. Since such testing is performed once per application version, the overhead is acceptable.
> >
> > **Data preparation for controlled text generation models:** When training models that must strictly follow complex constraints, we require a large amount of high-quality, constraint-compliant data. Our method allows developers or researchers to efficiently filter or curate training data at scale. In this setting, validation occurs offline during dataset construction, making the overhead reasonable for data quality assurance.
> >
> > Given these use cases, the computational cost of our method is justified, as it replaces substantial human labor and provides greater consistency and reliability.
> >
> > Thank you once more for your careful review and constructive suggestions!

---

### Official Review · Reviewer_do21 · 2025-11-01

**Soundness:** 3
**Presentation:** 3
**Contribution:** 3
**Rating:** 4
**Confidence:** 4

**Summary:**

This paper introduces ConstrainPrompt, a verification pipeline for code-verifiable, semantically agnostic constraints in LLM output. ConstrainPrompt induces code-verifiable constraints from natural language prompts, synthesizes a logical evaluation tree to determine order and scope, and compiles it into an executable verifier that returns a deterministic pass/fail decision with human-readable justification and provenance. On paired data of real-world prompts and model outputs, the proposed approach consistently outperforms a baseline judged by LLM, significantly improving constraint compliance accuracy and violation justification. Ablation studies confirm the critical role of the evaluation tree in both accuracy and explainability.

**Strengths:**

1. By introducing the evaluation tree, ConstrainPrompt can effectively separate global analysis and local checks, and respect the order from coarse to fine, reducing the common misjudgments and omissions of  LLM-as-a-judge.
2. ConstrainPrompt automates the extraction and compilation of natural-language constraints into executable code, enabling deterministic, reproducible validation that is immune to the inconsistency and subjectivity of human or LLM-based judges.
3. The method demonstrates significant and consistent improvements over LLM-as-a-judge across multiple state-of-the-art models, with gains of up to 39.5% in accuracy and 93.4% in violation rationale quality, underscoring its practical utility.

**Weaknesses:**

1. The pipeline heavily depends on powerful LLMs (like GPT-4o, Claude Sonnet) for constraint extraction, evaluation tree synthesis, and code generation. This raises concerns about the method's generalizability and accessibility. The paper does not demonstrate that the pipeline remains robust when using weaker models (e.g., smaller open-source models).
2. The core of this method is to only process "code-verifiable" constraints. However, this filtering step itself is judged by an LLM, which could become a source of error and a single point of failure. If the filter misclassifies a constraint (e.g., incorrectly judging a code-verifiable constraint as non-verifiable, or vice versa), the entire verification process becomes incomplete or inaccurate. There is a lack of deterministic guarantees for this critical step.
3. The paper primarily compares ConstrainPrompt against a simple “LLM-as-a-judge” baseline. However, a comparison with carefully engineered, hand-crafted rule-based validation systems would be more compelling. Such a comparison would more clearly measure the advantages and disadvantages of this automated approach in terms of accuracy and efficiency compared to human-expert-built, task-specific validators.

**Questions:**

1.Your approach relies on state-of-the-art models like GPT-4o or Claude Sonnet for code generation. Have you evaluated the performance degradation of various stages of your approach (particularly constraint extraction and code generation) when using less powerful (e.g., 7B-13B parameter size) open-source models like Llama or Qwen?

2.The core of this approach relies on a "code verifiability" filter determined by the LLM. Have you evaluated the accuracy of this filter itself? In your research, have you encountered cases where the entire verification process failed due to misjudgments by the filter (e.g., filtering out verifiable constraints or retaining unverifiable constraints)?

3.The paper compares ConstrainPrompt to a "LLM-as-a-judge" baseline and demonstrates significant improvement. Have you considered comparing your approach to hand-crafted, task-specific verifiers? Such a comparison would better illustrate your approach's accuracy advantage.

---

> ### Author Response · Authors · 2025-11-20
>
> Dear Reviewer do21,
>
> Thanks a lot for your constructive comments!
>
> Regarding the concern about generalizability of our method, we fully understand and agree that relying on strong LLMs may limit accessibility. To assess the robustness of our approach, we additionally conduct experiments on two weaker open-source models: **Qwen3 14B** and **Gemma3 12B**. The results are shown below:
>
> | Model       | Constraint Compliance Accuracy (LLM-as-a-judge) | Constraint Compliance Accuracy (Our method) | Violation Rationale (LLM-as-a-judge) | Violation Rationale (Our method) |
> |------------|--------------------------------------------------|----------------------------------------------|----------------------------------------|-----------------------------------|
> | **Qwen3 14B**  | **62.1**                                             | 59.1 (-4.8%)                                  | **0.4412**                                 | 0.3949 (-10.5%)                   |
> | **Gemma3 12B** | 53.0                                             | **74.2 (+40%)**                                   | 0.4061                                 | **0.4421 (+8.9%)**                    |
>
> We observe that our method improves both metrics on Gemma3 12B, while experiencing degradation on Qwen3 14B. After analyzing the data, we found that the main source of discrepancy comes from differences in code generation capability. Qwen3 14B has a **15.2% failure rate** in code generation (compilation errors or runtime errors), whereas Gemma3 12B has only **6.1%**. These results indicate that our approach can still benefit weaker models, but its performance is indeed sensitive to the model’s ability to generate valid code. Our method depends on the model’s basic code understanding to compensate for limitations in instruction-following, so models with insufficient code-related knowledge naturally lead to reduced effectiveness.
>
> We also fully acknowledge your concern that the correctness of the LLM-based code-verifiability filter significantly affects the overall pipeline. To better understand this, we randomly sample 20% of both successful and failed cases and manually examined each step. For successful cases, the accuracy of Step 1 **(prompt → code-verifiable constraints)** is **90.9%**, and the accuracy of Step 2 **(code-verifiable constraints → evaluation tree)** is **95.5%**. For failed cases, the accuracy drops to **27.3%** for Step 1 and **81.8%** for Step 2. This suggests that most failures indeed originate from misclassification of code verifiability, which clearly highlights a direction for future improvement. We sincerely appreciate this insightful remark.
>
> Finally, regarding rule-based baselines: task-specific rule-based validators typically require substantial manual design and only apply to narrowly defined tasks. For example, prior work such as [1] computes keyword coverage for a specific dataset, but such methods correspond to only **one fixed prompt template** in our setting. Since our goal is to handle **general and diverse prompts**, simple rule-based solutions cannot meaningfully scale or generalize in the way our method targets. Therefore such comparisons are not well aligned with the broader scope of our task.
>
> We sincerely thank you again for your thoughtful feedback, which has greatly helped us strengthen and clarify our work.
>
> [1] Carlsson, Fredrik, et al. "Fine-grained controllable text generation using non-residual prompting." Proceedings of the 60th Annual Meeting of the Association for Computational Linguistics (Volume 1: Long Papers). 2022.

---

### Official Review · Reviewer_n5wZ · 2025-11-09

**Soundness:** 3
**Presentation:** 4
**Contribution:** 3
**Rating:** 6
**Confidence:** 4

**Summary:**

The authors study prompts inside programmatic LLM applications, which tend to have hard constraints expressed in natural language. They introduce a method to turn natural language constraints into an executable verification function. Their method, ConstrainPrompt, starts by extracting a list of "code-verifiable" constraints (e.g., format checks, numerical checks, or the absence of certain tokens), each of which may possibly be conditioned on some criteria. Then, ConstrainPrompt sorts the constraints by scope from coarse-grained to increasingly fine-grained, dependent, or lower-level ones. The authors define this as a tree, though (as discussed below) it appears to me that this is just an ordered list of possibly conditional checks, with early exit on first failure.

The authors also introduce a dataset of real prompts from such LLM systems and, for a small number of them, collect corresponding LLM outputs and annotate them with violations of the hard constraints specified in the prompts. They use this data to investigate patterns of constraints and failures in real systems and the smaller subset of it to compare their method, ConstrainPrompt, against simply asking an LLM to judge model outputs against (hard constraints from?) the original prompts. Across three models, the authors observe the large gains in quality along two axes: accuracy of detecting compliance and attributing failures correctly.

**Strengths:**

This work explores an understudied problem: more carefully defining and evaluating the reliability of prompts inside programmatic LLM systems, particularly along the axis of explicit constraints in the prompts. It does so in a way that I think can help future work: the larger dataset, the taxonomy produced, and the smaller annotated data can be a sensible starting point for multiple future projects in this area.

The method introduced is simple and might be a good starting point for methods in this space, and the gains against LLM judges appear substantial, not to mention that code-based validators are likely cheaper and more interpretable than judges. Though the data and the scope of constraints are quite small, these types of constraints are nonetheless almost ubiquitous, so the problem studied can still have a reasonable amount of impact.

**Weaknesses:**

The scope of the study (only a handful of types of hard constraints) and the amount of data labeled for the evaluation (61 examples?) are alarmingly small. While I commend the authors for their transparency in describing their process, the filtering applied is substantial, e.g. keeping only "templates that contain only one user–input placeholder, which simplifies controlled input synthesis". It can be hard to ascertain how difficult all of this really is, especially as models and judges get better or the constraints become more complex.

The method described uses a "tree", but as described in the summary it appears that the nature of this binary tree is more simply characterized as just an (ordered) list of conditional checks, with early exit. While it is of course a valid tree, a simple list with "early exit" is perhaps better since it's simpler than the design space of "trees" could evoke.

The baselines are not necessarily most convincing. Were the judges "engineered" to align with the complete specification of the authors' intent from these evaluations? For example, the system is designed to prioritize certain types of violations over others (e.g., coarse-grained ones and focus on hard constraints). Is the LLM judge informed of all that? This matters for the Violation Rationale output and probably also for Constraint Compliance Accuracy. Why can't modern LLM judges check all these extremely simple constraints? Perhaps modern reasoning models, which are not particularly new anymore, can do this out of the box? The reason this concern matters is that it appears that the authors want to argue that their method is superior to simple judges, so the reasons for this argument need to be clearly argued or supported.

**Questions:**

See weaknesses.

---

> ### Author Response · Authors · 2025-11-20
>
> Dear Reviewer n5wZ,
>
> Thank you very much for your thoughtful and constructive review!
>
> Regarding the scope of our study: although our work currently focuses on a set of hard constraints, this set already reflects the most commonly used hard constraint types based on our empirical analysis. Our main motivation for targeting hard constraints is that, while LLMs are generally capable of handling semantic (soft) constraints due to their extensive pretraining on diverse text corpora, they still struggle with relatively simple non-semantic requirements such as counting or enforcing strict formats. This observation motivates us to use code-based logic to complement model weaknesses in these non-semantic areas. We fully understand your concern about the dataset size. Unfortunately, collecting this type of data is challenging: to the best of our knowledge, there is currently no publicly available large-scale dataset specifically designed for real-world prompt constraint compliance evaluation. As a result, we must manually examine each test case and label every generated output, which limits the scale of the dataset. Expanding the dataset to better reflect diverse real-world prompts is one of our key directions for future work. As for the filtering that restricts prompt templates to those containing only a single user-input placeholder: this choice is aligned with the setup of prior work on automatic test-case generation for prompt templates. Through our analysis, we observed that when LLMs generate inputs for templates with multiple interdependent placeholders, models often fail to capture cross-placeholder dependencies, for example, generating a full article and its summary as two unrelated pieces of text. To avoid introducing errors into the synthetic inputs, we thus constrained our dataset to single-placeholder templates in this version of the study.
>
> Regarding the “tree” structure: we would like to clarify that our evaluation tree is not merely a list of conditional checks with early exit. It covers both conditional and unconditional constraints, and it encodes the priority relationships among constraints (e.g., some checks must be handled before others because they serve as prerequisites). The tree representation therefore captures a hierarchical structure, not simply a linear sequence.
>
> For the LLM-as-a-judge baseline, we explicitly specify the definition of hard constraints and instructed the model to focus exclusively on those constraints. We agree that modern reasoning models may perform better than standard chat models when used as judges, since they offer stronger multi-step reasoning and more deliberate decision-making. However, their improved reasoning ability does not eliminate the need for our method. Reasoning models are still natural-language judges whose decisions remain nondeterministic, may vary across runs, and can still overlook subtle structural or numerical violations. In contrast, our method ultimately converts the verification process into deterministic, executable code, ensuring that every constraint can be checked reliably and reproducibly. Importantly, stronger reasoning models would not replace our method, instead, they would enhance it. Our pipeline relies on the model's instruction-following and analytical abilities in the earlier stages (such as code-verifiable constraint extraction and evaluation tree construction) before the final code-based checker is executed. A reasoning-oriented model with stronger comprehension and decomposition abilities would therefore improve these intermediate steps and make the entire pipeline even more robust. Thus, while reasoning models may improve the baseline judge to some extent, they do not eliminate the core problem our method addresses, and their strengths in fact make them complementary to our approach rather than a substitute.
>
> Thank you again for the valuable feedback.

---

### Meta-Review · Area_Chair_A4j3 · 2026-01-10

**Summary:**

This paper proposes generating code corresponding to hard constraints expressed in natural language in the prompt. Then the generate code is executed to determine if the constraints are satisfied. This is evaluated on a dataset proposed by the paper. We can accept simple method if it solves an important problem. For this paper the method is trivial, the problem is artificial, and the reviewers are not enthusiastic, so this paper should be rejected.

The reviewers raised the following points that are addressed to limited extend by the authors.

## Strengths

• **Novel problem focus** (n5wZ, oYjk): Addresses understudied area of validating prompt-defined constraints in LLM workflows

• **Automated code validation** (do21, ga62): Deterministic, reproducible validation immune to LLM inconsistency; extracts constraints into executable code

• **Strong performance** (do21, ga62): 39.5% accuracy and 93.4% violation rationale improvement over LLM-as-a-judge baseline

• **Evaluation tree design** (do21, ga62): Guard-first coarse-to-fine ordering enforces logical precedence, improves robustness

• **Practical advantages** (n5wZ): Code validators cheaper and more interpretable than LLM judges

---

## Weaknesses

• **Tiny evaluation scale** (n5wZ, ga62, oYjk): Only 61 annotated examples; single benchmark insufficient for real-world diversity

• **Strong LLM dependency** (do21, ga62): Requires GPT-4o/Claude; no evaluation with weaker models; generalizability and accessibility concerns (partly addressed, but also somewhat validate the reviewer concern that it does not work well with weaker models)

• **Weak baselines** (n5wZ, ga62): Missing comparisons with hand-crafted validators, constrained decoding, agentic workflows, reasoning models (not well addressed)

• **LLM filter is failure point** (do21, ga62): Code-verifiable filtering itself uses LLM, creating single point of failure with no deterministic guarantees

• **Limited constraint scope** (n5wZ, ga62): Only handles hard syntactic constraints; heavy filtering (single placeholder only) limits coverage

• **Questionable importance** (oYjk): Handwritten validators are one-time effort; ConstrainPrompt only handles easy syntactic checks

• **Strong models already good** (ga62): GPT-4.1 already aligns well with complex requirements; instruction-following improving

**Reviewer Concerns:**

mostly not addressed and cannot be addressed

**Reviewer Scores:**

probably not

---

### Decision · Program_Chairs · 2026-01-26

Reject